# Human-driven greenhouse gas and aerosol emissions cause distinct regional impacts on extreme fire weather

Danielle Touma [1✉], Samantha Stevenson[1], Flavio Lehner [2,3,4] & Sloan Coats[5]

Attribution studies have identified a robust anthropogenic fingerprint in increased 21st century wildfire risk. However, the risks associated with individual aspects of anthropogenic aerosol and greenhouse gases (GHG) emissions, biomass burning and land use/land cover change remain unknown. Here, we use new climate model large ensembles isolating these influences to show that GHG-driven increases in extreme fire weather conditions have been balanced by aerosol-driven cooling throughout the 20th century. This compensation is projected to disappear due to future reductions in aerosol emissions, causing unprecedented increases in extreme fire weather risk in the 21st century as GHGs continue to rise. Changes to temperature and relative humidity drive the largest shifts in extreme fire weather conditions; this is particularly apparent over the Amazon, where GHGs cause a seven-fold increase by 2080. Our results allow increased understanding of the interacting roles of anthropogenic stressors in altering the regional expression of future wildfire risk.

[1] Bren School of Environmental Science and Management, University of California, Santa Barbara, Santa Barbara, CA, USA. [2] Institute for Atmospheric and Climate Science, ETH Zürich, Zürich, Switzerland. [3] Climate and Global Dynamics Laboratory, National Center for Atmospheric Research, Boulder, CO, USA. [4] Department of Earth and Atmospheric Sciences, Cornell University, Ithaca, NY, USA. [5] Department of Earth Sciences, University of Hawai'i at Mānoa, Honolulu, HI, USA. ✉email: touma@ucsb.edu

Wildfire severity and area have increased significantly in recent decades over many regions[1–4]. To better understand these changes, global climate model experiments have been used to extract the human fingerprint on observed fire-weather conditions—warm, dry, and windy conditions that lead to dry fuels and fire spread. Compared to a world without anthropogenic climate change, higher temperatures and lower precipitation rates have caused extreme fire-weather conditions to become significantly more likely in the past four decades over the western US[5,6], causing recent fires in California to be unprecedented in severity and area[7,8]. In western Canada, anthropogenic forcing in 2011–2020 caused the risk of extreme fire-weather conditions to increase by up to a factor of six, substantially extending the fire season[9]. Likewise, anthropogenic forcing increased fire risk by 30% during the 2019–2020 Australian bushfires[10], which caused tens of fatalities and thousands of lost homes[11]. In Southern Europe, the extreme fire season witnessed in 2003 was 50 times more likely in today's climate compared to a world without anthropogenic climate change[12]. By mid-2020, the Siberian tundra had already experienced a more severe fire season than the previous record-breaking year under extreme heat conditions attributed to anthropogenic climate change, and the Amazon rainforest was predicted to exceed 2019's record number of observed fires.

Consistent with these recent increases in extreme fire weather, future anthropogenic climate change is expected to continue to increase the frequency and severity of wildfires. By the end of the 21st century, the frequency of extreme fire-weather conditions is projected to increase significantly over California[6,13], and more severe wildfires are expected in the western US as a whole[14]. In fact, for two-thirds of the global burnable area, extreme fire-weather conditions are expected to become the new normal by 2100[15]. The magnitude of these changes will of course depend on the future emissions scenario, but are a common feature even of low-emissions projections; for instance, in Sweden, a future global warming scenario of 2° C doubles the risk of extreme fire weather[16].

Understanding fire risk at a regional scale is important for mitigation and planning purposes. However, it is unclear how the competing influences of greenhouse gases, aerosols due to industrial and biomass burning activities, and land-use/land-cover change (LULC) have historically and will continue to modify regional fire-relevant climate conditions. In particular, the cooling effect of aerosols or changes in albedo due to LULC in mitigating the warming effect of greenhouse gases on extreme fire weather has yet to be quantified. In addition, greenhouse gases, aerosols, and LULC have distinct impacts on the spatiotemporal patterns of precipitation, relative humidity, and surface wind[17–21], but the effect of these changes on regional wildfire risk is still ambiguous.

Here, we use large ensemble simulations run with the Community Earth System Model version 1 (CESM1) capable of isolating individual climate forcings[22,23] to show that anthropogenic greenhouse gases, industrial and biomass burning aerosol emissions, and LULC have distinct influences on regional wildfire risk, both historically and under projected future climate change. We find that the risk of extreme fire-weather conditions has already doubled in the Amazon and increased by 50% in the Mediterranean, due to anthropogenic greenhouse gases. By the end of the 21st century, anthropogenic greenhouse gas-driven changes in temperature, relative humidity, and surface wind speed more than double the risk of extreme fire-weather conditions in the Amazon, eastern North America, the Mediterranean, southern Africa, and Southeast Asia. While industrial aerosols have historically reduced the risk of extreme fire weather conditions, their projected decline causes their effects to be diminished by 2080 in many fire-prone regions. Quantifying the regional expressions of anthropogenic forcing on extreme fire-weather risk could aid in climate change mitigation and adaptation strategies.

## Results

**Forced trends in extreme fire-weather risk.** We use the Canadian Forest Fire Weather Index (FWI)[24] to quantify weather conditions that enable fire ignition and spread. The FWI accounts for maximum temperature, precipitation, relative humidity, and surface wind to describe fuel moisture conditions on daily to monthly time scales. Extreme levels of FWI (exceeding the 95th percentile; see "Methods"—hereinafter extreme fire weather) indicate that the conditions can lead to high levels of fire severity and spread during a fire (e.g., refs. [6,9,15]). We quantified the probability of extreme fire weather over the fully forced (ALL) and all-but-one forced (X) Community Earth System Model version 1 Large Ensemble (CESM-LE and CESM-LE-SF, repsectively) simulations, and assessed the effect of aerosol emissions, greenhouse gas emissions, biomass burning emissions, and land-use/land-cover change (LULC) on extreme fire-weather risk over the historic and future periods (see "Methods").

While greenhouse gas emissions have had relatively small effects on extreme fire-weather risk prior to the mid-20th century (Supplementary Fig. 1), they have robustly increased extreme fire-weather risk in more recent decades (Figs. 1 and 2). Between 1980 and 2005, greenhouse gases have amplified the risk of extreme fire weather in Western and Eastern North America, the Mediterranean, Southeast Asia, and the Amazon by at least 20% (Figs. 1b and 2a–d, g). In the northeast of the Amazon region, the risk of extreme fire weather had already doubled under greenhouse gas emissions by 2005 (Fig. 1b). By 2080, greenhouse gases are expected to increase the risk of extreme fire weather by at least 50% in western North America, equatorial Africa, Southeast Asia, and Australia (Figs. 1h and 2a, e, g, h) and at least double this risk in the Mediterranean, southern Africa, eastern North America and the Amazon (Figs. 1h and 2b–d, f). Most notably, in parts of the Amazon, projected greenhouse gas emissions increase extreme fire-weather risk by >7 times in 2070–2080 (Figs. 1h and 2c). Taken together, these results indicate that greenhouse gas-driven changes in climate conditions have already increased the probability of extreme fire weather over many parts of the globe, and will further increase extreme fire-weather risk by the end of the 21st century.

Industrial aerosols, biomass burning aerosols, and LULC can either amplify or dampen greenhouse gas-driven increases of extreme fire-weather risk, and their effects are spatially variable. In the 20th century, industrial aerosols have reduced extreme fire-weather risk in the Amazon and Mediterranean by ~30%, but have amplified the risk by at least 10% in Southeast Asia and western North America (Figs. 1a and 2a, c, d, g and Supplementary Fig. 1). Unlike industrial aerosols, biomass burning aerosols have caused extreme fire-weather risk to increase by at least 30% over the Amazon and western North America during this same period (Supplementary Fig. 2). In addition, through steady increases in crop and pasture land over the 20th century[25], LULC has amplified extreme fire-weather risk in the Amazon and western Australia (Supplementary Fig. 2). However, we note that the land model component of CESM1 does not account for irrigation, which could also modify regional fire-weather related variables through land–atmosphere feedback, especially in extensive crop regions[26].

As decreases in industrial aerosols are projected through the 21st century, they no longer compensate for the severe greenhouse gas-driven increases in extreme fire-weather risk[27] (Fig. 1c, e, g and Supplementary Fig. 3). In fact, Northeastern Europe, the southern edge of the Amazon, western North America, and parts of Australia have a slight increase in the risk of extreme fire weather due to aerosol forcing by the end of the 21st century (Figs. 1c, e, g and 2a, h). In regions where industrial aerosols are still not reduced to pre-industrial levels in the future, such as the Horn of Africa, Central

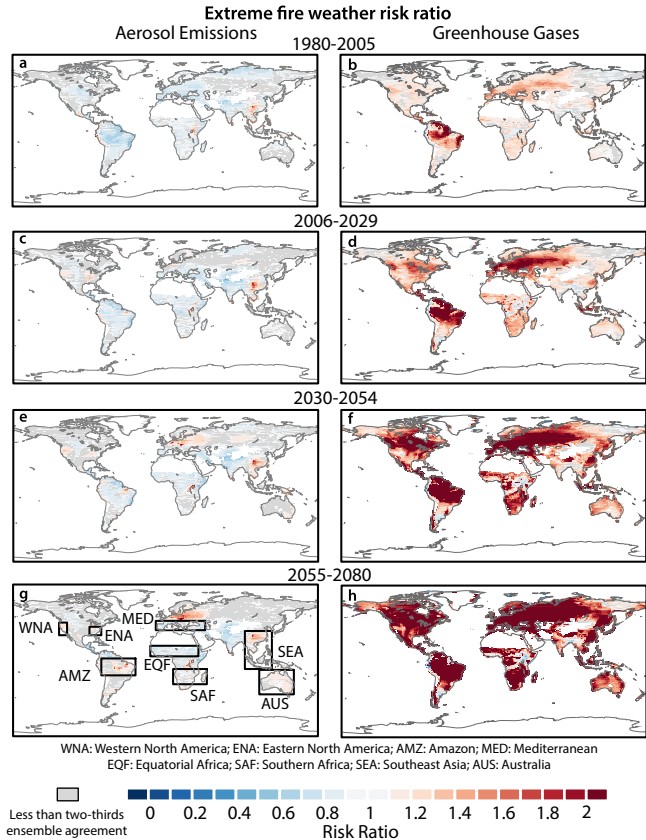

**Fig. 1 Risk ratio of extreme fire weather under different anthropogenic forcings.** Risk ratio (RR) for 1980–2005 (**a**, **b**), 2006–2029 (**c**, **d**), 2030–2054 (**e**, **f**), and 2055–2080 (**g**, **h**) under aerosol (AER) (**a**, **c**, **e**, **g**) and greenhouse gas (GHG) (**b**, **d**, **f**, **h**) forcing. RR for each period is calculated as the probability of exceeding the 95th percentile of the baseline daily fire-weather index (FWI) distribution in the all-forcing (ALL) ensemble divided by the probability of exceeding that same threshold in the all-but-one (X) forcing ensemble. The baseline FWI distribution for each period is the distribution of FWI in the ALL ensemble in that period. Regions in red (blue) have an increased (diminished) extreme fire-weather risk under a single forcing. Grid points masked in gray have less than two-thirds ensemble agreement on whether the RR is greater or less than one. Oceans, glaciers, and bare land are masked in white. The boxed areas define the regions used for the regional analysis are labeled in panel **g** and are described in Supplementary Table 2.

America, and northeastern Amazon, extreme fire-weather risk continues to be reduced (Figs. 1c, e, g and 2e, and Supplementary Fig. 3). However, any industrial aerosol-driven compensating effects in the 21st century are relatively small compared to the magnitude of greenhouse gas-driven increases in extreme fire-weather risk. Moreover, given that biomass burning aerosol emissions are not projected to decrease substantially throughout the 21st century under the high-emissions (RCP 8.5) scenario[27], and that deforestation is expected to continue over South America[25], extreme fire-weather risk will continue to be elevated above the greenhouse gas-driven increases over the Amazon and western North America (Fig. 2a, c and Supplementary Fig. 2).

**Mechanisms for forced trends in extreme fire-weather risk.** We next examine the relative roles of meteorological variables in contributing to changes in extreme fire-weather risk (maximum temperature, precipitation, relative humidity, and surface wind speed). To do so, we remove the forced signal for each variable individually from the ALL forcing simulations, and then

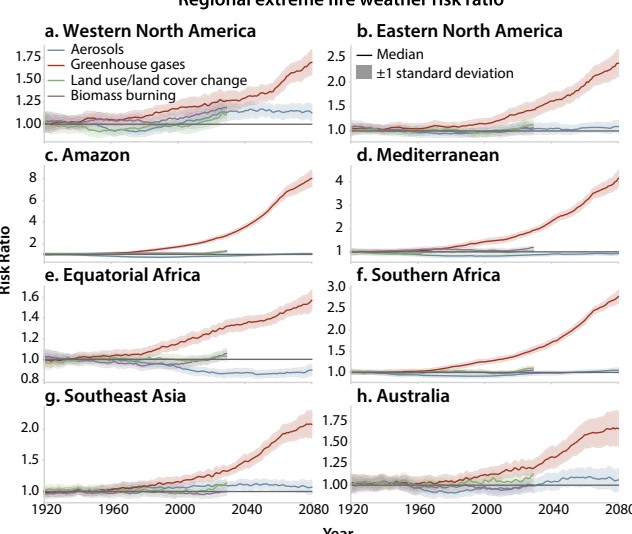

**Fig. 2 Regional extreme fire-weather risk ratio.** Regional time series of the 30-year moving risk ratio (RR) of extreme fire weather for different anthropogenic forcings. The 30-year RR is the regionally-averaged probability of exceeding the 95th percentile of the 30-year moving window daily fire-weather index (FWI) in the all-forcing (ALL) ensemble divided by the regionally-averaged probability of exceeding that same threshold in the all-but-one aerosol (blue), greenhouse gas (red), biomass burning (purple), and land-use/land-cover (green) X ensembles. The shaded envelope represents one standard deviation of the ensemble spread. Regional definitions are shown in Fig. 1h and Supplementary Table 2 and are **a** Western North America, **b** Eastern North America, **c** Amazon, **d** the Mediterranean, **e** Equatorial Africa, **f** Southern Africa, **g** Southeast Asia, and **h** Australia. All-but-one aerosol and greenhouse gas X simulations are run to 2080, while all-but-one biomass burning and land-use/land-cover X simulations are only run to 2029 (see "Methods").

recalculate extreme fire-weather risk ratios between the original and detrended ALL forcing ensembles (see "Methods"). This allows us to isolate the impact of each forcing on each variable, and provide insight into the mechanisms that drive extreme fire-weather risk.

The greenhouse gas-driven increase in daily maximum temperature is the dominant contributor to increases in extreme fire-weather risk over the globe; these effects amplify extreme fire-weather risk by at least 20% globally and by >80% in the Amazon, eastern North America, and Southeast Asia throughout the 21st century (Fig. 3b and Supplementary Figs. 4 and 5). In parts of the Amazon, decreases in relative humidity and increases in wind speed under greenhouse gas forcing (Supplementary Figs. 6 and 7) individually more than double the risk of extreme fire weather (Fig. 3f, h). This is consistent with our knowledge of fire dynamics, as both increases in wind speed and decreases in relative humidity can increase fire spread potential, while the latter can also cause severe losses in the underlying fuel moisture[28]. Aerosol-driven decreases in maximum daily temperatures compete with the warming effects of greenhouse gases in the 20th century to reduce extreme fire-weather risk in the Mediterranean, western North America, and parts of the Amazon (Supplementary Fig. 8). However, by 2080, the effect of aerosol emissions on maximum daily temperature dissipates over most of the globe and no longer mitigates increases in extreme fire-weather risk (Fig. 3a and Supplementary Figs. 5 and 8). Moreover, the continuation of aerosol-forced suppression of precipitation over Southeast Asia is projected to increase extreme fire-weather risk by at least 30% (Fig. 3c and Supplementary Figs. 8 and 9). While forced changes

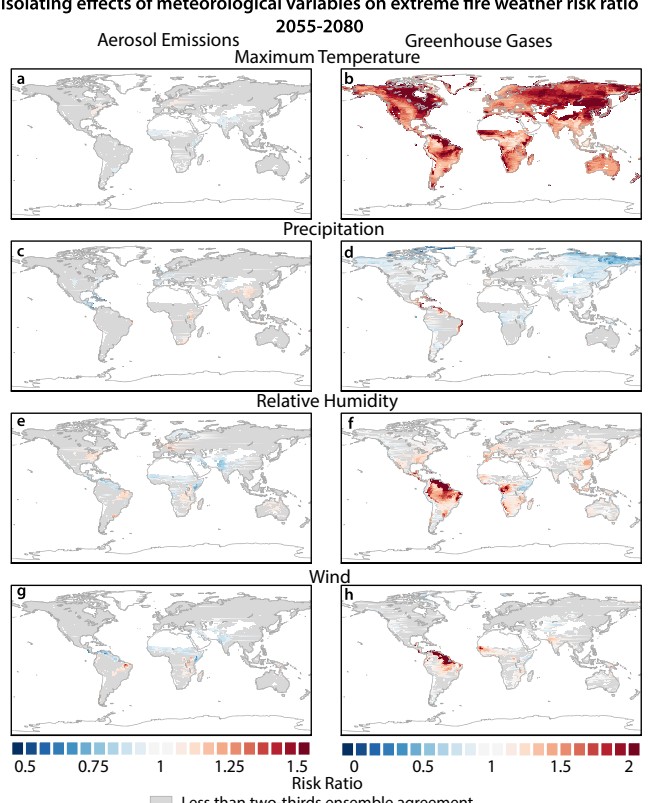

Isolating effects of meteorological variables on extreme fire weather risk ratio
2055-2080

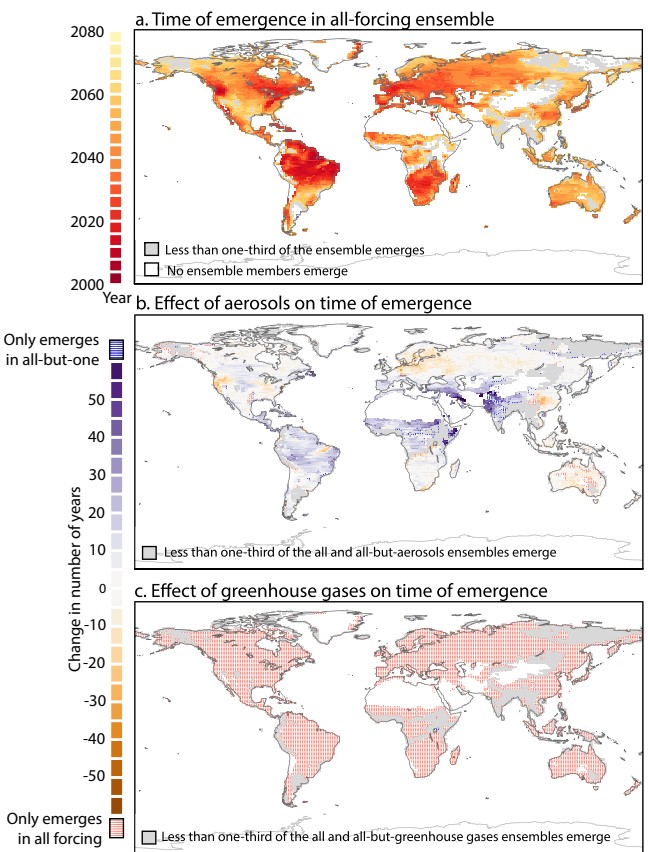

**Fig. 3 Isolating the effects of meteorological variables on extreme fire-weather risk.** The risk ratio (RR) is the probability of exceeding the 95th percentile of the baseline daily fire-weather index (FWI) distribution in the all-forcing (ALL) ensemble divided by the probability of exceeding that same threshold in the all-forcing ensemble after removing the aerosol (**a**, **c**, **e**, **g**) or greenhouse gas (**b**, **d**, **f**, **h**) effect on temperature (**a**, **b**), precipitation (**c**, **d**), relative humidity (**e**, **f**) and wind (**g**, **h**) in 2055–2080. The baseline FWI distribution for each period is the distribution of the FWI in the ALL ensemble in that period. Grid points masked in gray have less than two-thirds ensemble agreement on whether the RR is greater or less than one. Oceans, glaciers, and bare land are masked in white.

in global temperature cause robust and spatially homogeneous impacts on extreme fire-weather risk, the varied responses of wind, relative humidity, and precipitation result in more localized and nuanced effects on extreme fire-weather risk.

**The emergence of forced extreme fire-weather trends**. We have performed time of emergence (TOE) calculations to determine when forced changes to extreme fire weather emerge from the background of its historic variability; this occurs before 2080 for 74% of the global land area (Fig. 4a; see "Methods"), consistent with previous TOE studies using global climate models[15]. Over parts of the Amazon, eastern North America, the Mediterranean, and southern Africa, the frequency of extreme fire weather emerges beyond the historic variability as early as 2030 (Fig. 4a). Similar to a previous study using CMIP-class models, some parts of the Western US do not show emergence by the end of the century, despite robust projections of increased drought conditions for the 21st century[15,29]. In the Western US, warming of maximum daily temperature increases the risk of extreme fire weather under greenhouse gas emissions, while wetter atmospheric conditions dampen this risk, and impede the permanent emergence of extreme fire weather (Figs. 3 and 4a and Supplementary Figs. 5, 6, and 9).

**Fig. 4 Anthropogenic impact on the time of emergence (TOE) of extreme fire weather. a** The median TOE of extreme fire-weather frequency in the ALL forcing ensemble. TOE is the year at which the 10-year moving average of extreme fire-weather frequency exceeds and permanently remains above the baseline threshold, which is one standard deviation over the mean of the 1980–2005 all-forcing (ALL) ensemble. TOE is only shown for grid points with robust TOE, i.e., at least one-third of the ALL ensemble emerges before 2070. Gray areas show TOEs that are not robust, and white shading shows locations where none of the ensemble members emerge before 2070. **b** Median TOE difference between the ALL and the all-but-aerosol (XAER) ensemble. Negative (positive) values indicate an acceleration (delay) in the TOE. Blue stippling shows robust TOE for the XAER ensemble, but not for the ALL ensemble. Coral stippling shows robust TOE for the ALL ensemble, but not for the XAER ensemble. Gray areas have no robust TOE in both the ALL and XAER ensembles, **c**, **b** but for greenhouse gases (XGHG). Oceans, glaciers, and bare land are masked in white in all panels.

Using the CESM-LE-SF ensembles, it becomes possible to isolate the effects of each anthropogenic influence on the TOE. Greenhouse gases have caused a sharp acceleration in the TOE for regions with TOE before 2080; permanent emergence of extreme fire-weather frequency only occurs when including historic and future greenhouse gas forcing (i.e., does not occur when greenhouse gases are kept fixed at 1920 values; Fig. 4c). On the other hand, in parts of the Amazon, Mediterranean, and equatorial Africa, aerosol emissions have caused a 15-year delay in the TOE, partly mitigating the effects of greenhouse gases (Fig. 4b).

## Discussion
Our study provides novel insight into the pathways by which humans impact extreme fire weather. Anthropogenic greenhouse gas emissions have already doubled extreme fire-weather risk in some regions compared to a world with emissions fixed to pre-

industrial (1920) levels, and they will continue to increase extreme fire-weather risk throughout the 21st century. In fact, without mitigation of greenhouse gases, the frequency of extreme fire weather will emerge above its historic variability for much of the global burnable land area by 2080. While warming temperatures are largely responsible for increases in extreme fire-weather risk in most regions, greenhouse gas-driven decreases in relative humidity and increases in the surface wind over the Amazon will further amplify the risk. We cannot definitively attribute all mechanisms for these changes; however, the substantial drying over the Amazon is consistent with the southward shift of the Pacific ITCZ and weakening of the Atlantic Meridional Overturning Circulation (AMOC), and stronger surface winds are associated with increased land-ocean temperature gradients projected in future climates[30,31]. We note that changes in maximum temperature could itself influence relative humidity and surface wind, however, we do not account explicitly for these indirect effects in this study.

Aerosol-forced cooling substantially compensated for greenhouse gas-driven global warming in the 20th century, reducing extreme fire-weather risk and delaying the time of emergence. However, aerosols are projected to provide little to no relief for extreme fire-weather risk in the 21st century as they are reduced over most regions. These reductions drive warming and drying over Eastern North America and Europe, and amplify increases in extreme fire-weather risk. Over Southeast Asia, where aerosol emissions are projected to continue, the cooling effect of aerosols weakens the East Asian monsoon circulation, bringing drier conditions[32], and causing a net increase in extreme fire-weather conditions. Interestingly, biomass burning aerosols have an opposite effect to industrial aerosols in some regions. For example, the Amazon region experiences an increase in extreme fire-weather risk in the early 21st century due to biomass burning, possibly linked to inhibited cloud formation, which reduces low-intensity precipitation and increases temperatures[33,34]. These findings establish a more nuanced characterization of the anthropogenic impact on extreme fire-weather risk to inform mitigation and adaptation strategies.

Our study is the first to quantify the competing anthropogenic influences on extreme fire-weather risk in the historic and future periods. The large ensemble sizes of the CESM-LE and CESM-LE-SF datasets allowed us to quantify the range of natural climate variability relevant to fire-weather explicitly, without the need for statistical modeling or bootstrapping approaches, and robustly extract the anthropogenically forced response of extreme fire-weather risk. This adds to previous studies that have successfully disentangled the roles of greenhouse gases and aerosols on extreme heat[35] and precipitation events[36] using relatively large single-forcing ensembles. We note that our findings rely on a single climate model, which has structural uncertainties in its representation of the mean and variability of regional and global climate[37]. While the fully forced CESM-LE experiment is able to reasonably capture the spatial patterns of historic FWI compared to observations and other CMIP5 model simulations (Supplementary Figs. 10 and 11), analyzing additional single-forcing ensembles as they become available will help quantify structural uncertainty around the drivers of historic and future extreme fire-weather risk.

The present work focuses on the effect of greenhouse gas emissions, aerosols, biomass burning, and LULC on extreme fire-weather risk. We find that LULC imposes highly localized and small changes on extreme fire-weather conditions over the globe, though our findings are subject to the relatively small ensemble size and simulation length of the all-but-LULC experiment. In addition, changes to fuel sources and amounts, through land management and urban growth, and ignition rates, through the

development of wildland–urban interfaces and changes in lightning frequency, significantly alter wildfire occurrence and spread in historic and future climates[38–41]. However, global climate models are still limited in their representation of fire ignition and spread, as well as the effects of fire on vegetation. When included, these sub-grid-scale processes are highly parameterized in space and time and rely on limited observations[42]. Our study does not incorporate changes in ignition and vegetation, yet it sheds light on the future fire-weather conditions in which these changes could occur. In fire-prone regions that are not fuel-limited, such as boreal North America and Canada, Southeast Asia, and the Amazon[39,43], the increase in extreme fire-weather risk presented herein is especially significant. In addition, changes in climate and land management have already shifted some regional fire regimes from fuel to drought-limited, such as in the Mediterranean[44] and California[45], increasing the role of fire weather in fire spread and risk.

The findings of our study are key for understanding the impacts of climate change mitigation efforts on extreme fire-weather risk. The global footprints of greenhouse gases, aerosol emissions, LULC, and biomass burning through time can have implications for extreme fire-weather risk. Specifically, the interplay of the global influence of greenhouse gases on temperature with relatively localized and regional impacts of industrial and biomass burning aerosol emissions and LULC on temperature, relative humidity, wind, and precipitation can result in remarkable increases in extreme fire-weather risk. This may have implications for future wildfire management efforts. For example, the need for international coordination of firefighting personnel and aircraft assistance will increase[46] as large parts of the globe emerge into a new normal of extreme fire weather under increasing greenhouse gas emissions and reductions in aerosols.

## Methods

**All and all-but-one forcing large ensemble experiments.** We use the Community Earth System Model version 1 (CESM1) Large Ensemble (CESM-LE) historical and Representative Concentration Pathway (RCP) 8.5 fully forced (ALL) simulations[47]. The CESM1 model, which is used for all CESM-LE simulations, couples the Community Atmospheric Model version 5 (CAM5), Parallel Ocean Program, version 2 (POP) model, Community Land Model, version 4 (CLM4), and Los Alamos Sea Ice Model (CICE) component models. A detailed description of the model components and physics can be found in Hurrell et al.[37]. We use simulations from the first 35 ALL forcing CESM-LE ensemble members. The first member was run from 1850 to 2100 under historical and RCP 8.5 forcing. The other ALL ensemble members branch out from the first member, starting in 1920 and extending to 2100 under the same forcings, but differ by a small random perturbation to their initial air temperature field in 1920 (see ref. [47] for more details). These simulations have been used extensively to quantify the impact of anthropogenic climate change on extreme climate events[48–51].

Large ensembles of climate simulations are useful testbeds for identifying forced changes to wildfire risk due to the statistical power they provide[5,9]. However, to date, few large ensembles isolating individual anthropogenic forcings have been created; the single-forcing Canadian Earth System Model (CanESM2)[52] and Community Earth System Model (CESM-LE-SF)[22,23] Large Ensembles are two notable exceptions. In addition to the 35 fully forced simulations, we use the newly generated CESM-LE-SF simulations[22]. In these simulations, greenhouse gases (XGHG), industrial (energy, transport, and residential) aerosols (XAER), aerosols from biomass burning (agriculture and wildfires; XBMB), and land-use/land-cover change (XLULC) are held constant at 1920 levels through the historical and RCP 8.5 scenarios[22]. Industrial (AER) and biomass burning (BMB) aerosol emissions differ in their sources, compositions, and historic and RCP 8.5 trajectories. AER emissions have risen steadily from 1860 until the present, and are projected to decrease steadily to pre-industrial levels by 2100[27]. BMB emissions have risen steeply from 1960 to the present, and while they are projected to decrease through the 21st century, BMB emissions will remain higher than pre-industrial levels[27]. The XAER and XGHG ensembles each contain 20 members; XBMB is a 15-member ensemble, and XLULC is a 5-member ensemble. The XAER and XGHG ensembles extend through 2080, while the XBMB and XLULC simulations end in 2029. The prescribed forcing of BMB in the RCP 8.5 scenario does not vary greatly from 2030 onwards, which partly motivated a cost-saving, shorter XBMB simulation ending in 2029[22,27]. The X ensemble members also branch out from the first ensemble member of the ALL simulations in 1920 by slightly perturbing the air

temperature field, but with fixed GHG, AER, BMB, and LULC forcing[22]. ALL and X ensemble output is available from http://www.cesm.ucar.edu/experiments/cesm1.1/LE/#single-forcing and http://www.cesm.ucar.edu/projects/community-projects/LENS. The ALL and X experiments are summarized in Supplementary Fig. 12.

**Fire-weather index**. We calculate the daily Canadian Forest Fire Weather Index (FWI) for each simulation using daily resolution output for maximum temperature (TREFHTMX, T hereafter), precipitation (PRECT, PR hereafter), relative humidity (RH), and average surface wind speed (WSPDSRFAV, WS hereafter)[24]. RH was calculated using the *MetPy* Python package using sea level pressure (PSL), lowest water vapor-mixing ratio (QBOT), and temperature (TREFHT)[53]. Using these daily variables, three fuel moisture codes are calculated to capture daily fuel moisture on the surface and in shallow and moderate depths of the soil layer[28]. Fire behavior is then quantified through two indices that combine the three fuel moisture codes and wind speed. These two initial spread and build-up indices are combined to create daily FWI. The FWI is unitless and theoretically ranges from 0 to ∞ but remains below 180 in most locations[54].

The FWI has been used to quantify fire conditions in previous studies using CESM simulations[10,16], and details for its calculation are described extensively by Wagner[24] and Dowdy et al.[28]. There are some discrepancies between FWI for CESM-LE over 1980-2018 compared to ERA-Interim reanalysis, used in lieu of observations. For instance, while the spatial patterns of the 95th percentile of FWI, which is used for the analyses herein, are similar, the range in magnitudes is narrower in the CESM-LE (Supplementary Fig. 10). These discrepancies, however, are relatively small. In addition, the spatial patterns and magnitude of the 95th percentile of the FWI for CESM-LE are comparable to those of other models from the fifth phase of the Coupled Model Intercomparison Project[55] (CMIP5; Supplementary Fig. 11). For this CMIP5 comparison, we used the first ensemble member that had all necessary variables (tasmax, pr, rhs, and sfcWind) available for each model (Supplementary Table 1).

**Anthropogenic impact on extreme fire-weather risk**. For each grid point, we calculate the 95th percentile of the daily FWI across all 35 members of the ALL ensemble for every 30-year moving window from 1920 to 2080 ($p95_t$). $P95_t$ is the threshold used to define extreme fire weather for each window, $t$. We calculate the probability of FWI exceedance above this threshold for each member, $i$, of the ALL ensemble,

$$P_{i,t}^{\text{ALL}} = P(\mathbf{FWI}_{i,t}^{\text{ALL}} \geq p95_t), \tag{1}$$

and each member, $j$, of the X ensemble,

$$P_{j,t}^{\text{X}} = P(\mathbf{FWI}_{j,t}^{\text{X}} \geq p95_t), \tag{2}$$

for each 30-year moving window, $t$. We take the ensemble-mean probability of the X ensemble, $\overline{P_{j,t}^{\text{X}}}$, for each 30-year moving window, $t$. We then calculate the risk ratio, $RR_{i,t}$, for each member, $i$, in the ALL ensemble for each 30-year moving window, $t$, to quantify the impact of each anthropogenic forcing on extreme fire-weather risk for each grid point:,

$$RR_{i,t} = \frac{P_{i,t}^{\text{ALL}}}{\overline{P_{j,t}^{\text{X}}}}. \tag{3}$$

In addition, for each location, we assess if at least two-thirds ensemble agreement on whether the RR is greater or less than one.

The risk ratio has been used widely in the detection and attribution climate change literature to understand the impact of anthropogenic forcing on the probability of extreme climate events (e.g., refs. [9,56–59]). A risk ratio greater than 1 indicates that anthropogenic forcing indicates an increase in the risk of extreme fire weather, while a risk ratio of less than 1 indicates a decrease. To summarize our results, we use six distinct periods over the historic and future periods: 1920–1949, 1950–1979, 1980–2005, 2006–2029, 2030–2054, and 2055–2080 (we do not summarize XLULC and XBMB over the last two periods). In addition, we use eight regions that have had significant burned area and fire occurrence over the past two decades to investigate regional variations[39]. Supplementary Table 2 summarizes the boundaries of these eight regions.

**Isolating the contribution of climate variables to extreme fire-weather risk**. First, we calculate the effect of AER, GHG, BMB, and LULC forcings on the individual climate variables used to calculate the FWI; namely, daily maximum temperature (T), precipitation (PR), relative humidity (RH), and surface wind speed (WS). For temperature, we calculate the forced effect via the following decomposition:,

$$\mathbf{T}_{\text{effect},i}^{\text{X}} = \mathbf{T}_i^{\text{ALL}} - \overline{\mathbf{T}_j^{\text{X}}}, \tag{4}$$

where $\mathbf{T}_{\text{effect},i}^{\text{X}}$ is the time-varying effect of anthropogenic forcing, X, on temperature for ensemble member $i$, $\mathbf{T}_i^{\text{ALL}}$ is the ALL forced temperature of ensemble member $i$, and $\overline{\mathbf{T}_j^{\text{X}}}$ is the time-varying ensemble mean of temperature over the X ensemble. Therefore, $\mathbf{T}_{\text{effect},i}^{\text{X}}$ is the forced response of temperature due to the anthropogenic

activity for ensemble member, $i$. The time-varying effects of P, RH, and WS are also calculated using Eq. (4). This approach is equivalent to that taken by Deser et al.[22] to compute the effects of individual forcings.

Then, for each ALL ensemble member we remove the forced response of T, as follows:,

$$\mathbf{T}_{\text{detrended},i}^{\text{ALL,X}} = \mathbf{T}_i^{\text{ALL}} - \mathbf{T}_{\text{effect},i}^{\text{X}} \tag{5}$$

For PR, RH, and WS, we ensure the detrended variable remains non-negative, and remove the forced response as a fractional effect, as follows:,

$$\mathbf{PR}_{\text{detrended},i}^{\text{ALL,X}} = \mathbf{PR}_i^{\text{ALL}} * \left(1 - \frac{\mathbf{PR}_{\text{effect},i}^{\text{X}}}{\overline{\mathbf{PR}_j^{\text{X}}}}\right) \tag{6}$$

We then recalculate the FWI for each detrended variable (T, PR, RH, WS) and for each anthropogenic forcing (X). Lastly, the risk ratio is recalculated to quantify the contribution of each variable to extreme fire-weather risk under anthropogenic forcing. In this case, we divide the probability of exceedance in the ALL forcing ensemble by the mean probability of exceedance in the detrended ALL forcing ensemble.

**Time of emergence**. Similar to the previous studies[15,29,49], we quantify the time of emergence (TOE) for each ensemble member of the ALL, XGHG, and XAER experiments as the year that the forced signal (10-year moving average) of the annual frequency of extreme fire weather exceeds and stays outside the historic variability. The historic variability is the mean and standard deviation across the ALL ensemble 10-year moving average extreme fire-weather frequency from 1980 to 2005. If TOE does not occur before 2070, we assume that the signal does not permanently emerge before 2080 (i.e., emergence has to occur for at least 10 years). In addition, we consider the TOE robust at a given location only when the signal permanently emerges for at least a third of the ensemble members. Then, we quantify the delay or acceleration of the TOE under greenhouse gases and aerosols by taking the difference between the median TOE of the ALL forcing ensemble and the median TOE of the respective X forcing ensemble.

## Data availability
The CESM1 Large Ensemble and the CESM1 Single Forcing Large Ensemble daily TREFHTMX, PRECT, PSL, QBOT, TREFHT, and WSPDSRFAV data are available at https://www.earthsystemgrid.org/dataset/ucar.cgd.ccsm4.CESM_CAM5_BGC_LE.atm.proc.daily_ave.html. The Fire Weather Index - ERA-Interim data are available at https://zenodo.org/record/1065401#.X6172pNKi3c. The daily tasmax, pr, rhs, and sfcWind daily data are available publicly through the Earth System Grid Federation at the following link: https://esgf-node.llnl.gov/projects/esgf-llnl/.

## Code availability
Figures 1–4 were created using python scripts and source data available publicly at https://github.com/detouma/FWI-figures.

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

## Acknowledgements

F.L. has been supported by the Swiss NSF (grant no. PZ00P2_174128), the NSF Division of Atmospheric and Geospace Sciences (grant no. AGS-0856145, Amendment 87), and the Regional and Global Model Analysis (RGMA) component of the Earth and Environmental System Modeling Program of the U.S. Department of Energy's Office of Biological & Environmental Research (BER) via NSF IA 1844590. We thank the CESM Project and CISL supercomputing resources (doi:10.5065/D6RX99HX) for providing access to the CESM-LE and CESM-SF-LE simulations and computing resources for the analysis. This is publication No. 11218 of the School of Ocean and Earth Science and Technology (SOEST).

## Author contributions

D.T. and S.S. contributed to the conception and design of the work. D.T. contributed to the acquisition, analysis, and interpretation of the data, and writing the first draft and revisions of the paper. S.S., F.L., and S.C. contributed to the interpretation of the data and substantial revisions to the paper.

## Competing interests

The authors declare no competing interests.

**Additional information**

