## [Peer Review File · Nature Communications]

Reviewer Comments, first round -

Reviewer #1 (Remarks to the Author):

This paper examines the role of different anthropogenic forcings (GHG, aerosols, biomass burning and land use/land cover) on global fire weather conditions over the historical and future periods under the RCP8.5 scenario. Authors used a large ensemble from the CESM-LE-SF model and disentangled the relative contribution of each of these factors on simulated changes to each meteorological variable shaping the fire weather index.

Authors show that GHG is the main driver of historical and future changes to extreme fire weather conditions but other forcings, especially aerosols have amplified or dampened the signal across parts of the world, delaying or accelerating the time of emergence of the fire weather index.

I enjoyed reading this study. This is a very nice paper, well written and well structured. To my knowledge, this is the first work disentangling the role of GHG, aerosols, biomass burning and LULC on future changes to fire weather. I think the methodology is robust and although the results rely on one single model, the conclusions are likely to apply to other CMIP5 or CMIP6 models. Finally, the results should be of interest to a broad audience so I think the paper should be published in Nature Communications.

All my comments below are minor.

Title - not sure that 'risk' is the best wording as it usually refers to the interplay between hazard, exposure and vulnerability. 'Fire weather conditions' or something would be more appropriate.

First paragraph of the introduction – an attribution exercise on fire weather conditions has also been performed recently in southern Europe.

Barbero, R., Abatzoglou, J. T., Pimont, F., Ruffault, J., & Curt, T. (2020). Attributing Increases in Fire Weather to Anthropogenic Climate Change Over France. *Frontiers in Earth Science*, 8(April), 1–11. <https://doi.org/10.3389/feart.2020.00104>

Ayres (2019), Khurshudyan (2020), Ciavarella et al. (2020) and Batista (2020) are not peer-reviewed papers and rather belong to the gray literature. I would suggest removing these references, especially in a Nature journal.

Line 61 – below is another paper that supports this statement

Williams, A. P., Abatzoglou, J. T., Gershunov, A., Guzman-Morales, J., Bishop, D. A., Balch, J. K., & Lettenmaier, D. P. (2019). Observed impacts of anthropogenic climate change on wildfire in California. *Earth's Future*, 2019EF001210. <https://doi.org/10.1029/2019EF001210>

Line 163-173 – why going back to Figure 1? This disrupts the flow.

Line 203 – could you briefly detail the mechanism responsible for that increase in wind speed over the Amazon?

Line 230 – Changes in fuel amount are also of primary importance as the climate-fire relationship has been shown to be function of the productivity gradient.

Line 229-242 – also, it would be nice to discuss the (contrasting) role of LULC on fire weather conditions vs actual fire activity.

Figure 2 – please indicate in the caption why some simulations end before 2080.

Figure 3 – both left and right panels provide (almost) the same information. Wouldn't it be more interesting to show the GHG forcing for say 2055-2080 on the left and the aerosols forcing on the right for the same time period?

Reviewer #2 (Remarks to the Author):

This is an excellent and novel study that utilizes climate model large ensembles, with and without a variety of anthropogenic forcings, to attribute projected changes in wildfire risk. The conclusions are interesting and useful, highlighting in particular the role aerosols have played in a number of regions in suppressing wildfire risk, and demonstrating impact of the future decline in that suppression effect. This is an important study, and I recommend acceptance subject to the authors addressing a few minor comments, primarily providing a few more details for the broad audience this paper will be of interest to.

Specific comments:

Line 93: The paper talks about "extreme" levels of wildfire risk, and chooses the 90th percentile of Canadian FWI as a criteria to define "extreme" fire behaviour. However, at a 90th percentile, on average that mean 36 days per year, one in ten days, would exceed this threshold. A 90th percentile threshold certainly describes weather suitable for fire spread, but events described as "extreme" might be expected to occur on fewer than 1 in 10 days a year. Other studies have utilized more stringent fire weather thresholds of 95 or 99% eg. see:

de Jong, M. C., Wooster, M. J., & McCall, F. F. (2016). Calibration and evaluation of the Canadian Forest Fire Weather Index (FWI) System for improved wildland fire danger rating in the United Kingdom. *Natural Hazards and Earth System Sciences*, 16(5), 1217.

Dowdy, A. J., Mills, G. A., Finkele, K., & de Groot, W. (2009). Australian fire weather as represented by the McArthur forest fire danger index and the Canadian forest fire weather index. *Centre for Australian Weather and Climate Research Tech. Rep*, 10, 91.

Arca, B., Pellizzaro, G., Duce, P., Salis, M., Bacciu, V., Spano, D., ... & Finney, M. (2010). Climate change impact on fire probability and severity in Mediterranean areas. In In: Viegas, DX, ed. *Proceedings of the VI International Conference on Forest Fire Research*; 15-18 November 2010; Coimbra, Portugal. Coimbra, Portugal: University of Coimbra.

Fréjaville, T., & Curt, T. (2015). Spatiotemporal patterns of changes in fire regime and climate: defining the pyroclimates of south-eastern France (Mediterranean Basin). *Climatic Change*, 129(1-2), 239-251.

I'm not suggesting the authors repeat the analysis with a stricter FWI threshold, rather that their analysis reflects more high fire danger conditions, rather than "extreme".

Line 119-124: This paper will attract broad interest, it would be useful if, in a number of places, the authors provide more explanation or relevant citations of the mechanisms behind the various forcings. In this section, aerosols are described as increasing fire risk in some areas, but decreasing in others, without clear discussion of the different mechanisms acting in each case. Similarly in lines 168-169; aerosols are described as suppressing precipitation, assuming the reader understands the meteorological process by which that takes place - why would aerosols be expected to decrease precipitation in this case and not decrease it? Useful reference may be:

Alizadeh-Choobari, O. (2018). Impact of aerosol number concentration on precipitation under different precipitation rates. *Meteorological Applications*, 25(4), 596-605.

Line 411: There is an error in this reference, the title is repeated twice.

- Reviewer: Dr Grant J Williamson

Response to Reviewers Document for “Competing anthropogenic effects cause distinct regional impacts on extreme wildfire risk” by Touma et al. (NCOMMS-20-32658-T)

We thank the Editor and Reviewers for their insightful comments. We have made constructive changes to the manuscript in response to each of these comments, and believe that the manuscript is substantially improved as a result of the changes.

This Response to the Reviewers file provides a complete documentation of the changes that are responsive to each individual Reviewer comment. The document is designed so that the changes that we have made in response to each comment can be immediately read and understood, independent of the other comments and responses. While this comprehensive comment-by-comment explanation requires some duplication of material throughout the document, our intention is that it helps to easily and efficiently evaluate exactly how each individual comment has been addressed.

Reviewer comments are shown in **bold**. Author responses are shown in plain text. Quotations from the revised manuscript are shown in *italics*.

REVIEWER COMMENTS

Reviewer #1 (Remarks to the Author):

This paper examines the role of different anthropogenic forcings (GHG, aerosols, biomass burning and land use/land cover) on global fire weather conditions over the historical and future periods under the RCP8.5 scenario. Authors used a large ensemble from the CESM-LE-SF model and disentangled the relative contribution of each of these factors on simulated changes to each meteorological variable shaping the fire weather index.

Authors show that GHG is the main driver of historical and future changes to extreme fire weather conditions but other forcings, especially aerosols have amplified or dampened the signal across parts of the world, delaying or accelerating the time of emergence of the fire weather index.

I enjoyed reading this study. This is a very nice paper, well written and well structured. To my knowledge, this is the first work disentangling the role of GHG, aerosols, biomass burning and LULC on future changes to fire weather. I think the methodology is robust and although the results rely on one single model, the conclusions are likely to apply to other CMIP5 or CMIP6 models. Finally, the results should be of interest to a broad audience so I think the paper should be published in Nature Communications.

We thank the reviewer for the positive comments on our manuscript.

All my comments below are minor.

Title - not sure that ‘risk’ is the best wording as it usually refers to the interplay between

hazard, exposure and vulnerability. ‘Fire weather conditions’ or something would be more appropriate.

Thank you for pointing this out. In the title, we are referring to the specific risk calculation we employ in the methods. However, given the broad readership of Nature Communications, we have modified the title to be more specific to fire weather conditions, as recommended by the reviewer. The new title is:

“Competing anthropogenic effects cause distinct regional impacts on extreme fire weather conditions.” (Lines 1-2)

First paragraph of the introduction – an attribution exercise on fire weather conditions has also been performed recently in southern Europe.

Barbero, R., Abatzoglou, J. T., Pimont, F., Ruffault, J., & Curt, T. (2020). Attributing Increases in Fire Weather to Anthropogenic Climate Change Over France. *Frontiers in Earth Science*, 8(April), 1–11. <https://doi.org/10.3389/feart.2020.00104>

Thank you for pointing this significant paper out to us. We have added a statement and citation in the introduction:

“In Southern Europe, the extreme fire season witnessed in 2003 was 50 times more likely in today’s climate compared to a world without anthropogenic climate change¹².” (Lines 50-52)

Ayres (2019), Khurshudyan (2020), Ciavarella et al. (2020) and Batista (2020) are not peer-reviewed papers and rather belong to the gray literature. I would suggest removing these references, especially in a Nature journal.

As suggested by the reviewer, we have removed these non-peer reviewed references.

Line 61 – below is another paper that supports this statement

Williams, A. P., Abatzoglou, J. T., Gershunov, A., Guzman-Morales, J., Bishop, D. A., Balch, J. K., & Lettenmaier, D. P. (2019). Observed impacts of anthropogenic climate change on wildfire in California. *Earth’s Future*, 2019EF001210. <https://doi.org/10.1029/2019EF001210>

Thank you for pointing this out. We have added this reference to the statement. (reference 13, Line 59)

Line 163-173 – why going back to Figure 1? This disrupts the flow.

We have removed the reference to Figure 1. Additionally, with the suggestion of your comment regarding Figure 3, we have modified Figure 3 to include the isolated variable effects of aerosols in 2055-2080, and now refer to specific panels to support relative statements:

“However, by 2080, the effect of aerosol emissions on maximum daily temperature dissipates over most of the globe, and no longer mitigates increases in extreme fire weather risk (Figure 3a, Supplementary Figures 5 and 8). Moreover, the continuation of aerosol-forced suppression of precipitation over Southeast Asia is projected to increase extreme fire weather risk by at least 30% (Figure 3c, Supplementary Figures 8 and 9).” (Lines 160-164)

Line 203 – could you briefly detail the mechanism responsible for that increase in wind speed over the Amazon?

We added text to explain the mechanisms that are projected to increase drying and wind speed over the Amazon:

“While warming temperatures are largely responsible for increases in extreme fire weather risk in most regions, greenhouse gas-driven decreases in relative humidity and increases in surface wind over the Amazon will further amplify the risk. We cannot definitively attribute all mechanisms for these changes, however, the substantial drying over the Amazon is consistent with the southward shift of the Pacific ITCZ and weakening of the Atlantic Meridional Overturning Circulation (AMOC), and stronger surface winds are associated with increased land-ocean temperature gradients projected in future climates^{31,32}.” (Lines 194-201)

Line 230 – Changes in fuel amount are also of primary importance as the climate-fire relationship has been shown to be function of the productivity gradient.

We thank the reviewer for pointing this out. We have added the important role of fuel amount in the text.

“Additionally, changes to fuel sources and amounts, through land management and urban growth, and ignition rates, through the development of “wildland-urban interfaces” and changes in lightning frequency, significantly alter wildfire occurrence and spread in historic and future climates³⁹⁻⁴².” (Lines 235-238)

Line 229-242 – also, it would be nice to discuss the (contrasting) role of LULC on fire weather conditions vs actual fire activity.

We have added the following statement to discuss the contrasting roles of LULC on fire weather conditions and actual fire activity.

“We find that LULC imposes highly localized and small changes on extreme fire weather conditions over the globe, though our findings are subject to the relatively small ensemble size and simulation length of the all-but-LULC experiment. Additionally, changes to fuel sources and amounts, through land management and urban growth, and ignition rates, through the development of “wildland-urban interfaces” and changes in lightning frequency, significantly alter wildfire occurrence and spread in historic and future³⁹⁻⁴².” (Lines 232-238)

Figure 2 – please indicate in the caption why some simulations end before 2080.

We have added the following statement in the caption to clarify that all-but-one biomass burning and land use/land cover X simulations are only run to 2029:

“All-but-one aerosol and greenhouse gas X simulations are run to 2080, while all-but-one biomass burning and land use/land cover X simulations are only run to 2029 (see Methods).”
(Lines 591-593)

Additionally, in the Methods, we have explained the reason behind the decision that Deser et al. (2020) made to end the XBMB simulations in 2029:

“The prescribed forcing of BMB in the RCP 8.5 scenario does not vary greatly from 2030 onwards, which partly motivated a cost-saving, shorter XBMB simulation ending in 2029^{23,28}.”
(Lines 285-286)

Figure 3 – both left and right panels provide (almost) the same information. Wouldn't it be more interesting to show the GHG forcing for say 2055-2080 on the left and the aerosols forcing on the right for the same time period?

We thank the reviewer for this helpful comment. We have modified Figure 3 to include the isolated impacts of aerosols and greenhouse gases on each FWI variable for 2055-2080. This has improved the flow of the manuscript and shows information which we refer to in the text. We have moved the isolated impacts of greenhouse gases on each FWI variable for 2030-2054 to Supplementary Figure 4. The new caption for Figure 3 reads:

“Isolating the effects of meteorological variables on extreme fire weather risk under aerosol emissions (AER) and greenhouse gas (GHG) forcing. The risk ratio (RR) is the probability of exceeding the 90th percentile of the baseline daily FWI distribution in the all-forcing (ALL) ensemble divided by the probability of exceeding that same threshold in the all-forcing ensemble after removing the AER (a,c,e,g) or GHG (b,d,f,h) effect on temperature (a,b), precipitation (c,d), relative humidity (e,f) and wind (g,h) in 2055-2080. The baseline FWI distribution for each period is the distribution of the FWI in the ALL ensemble in that period. Grid points masked in grey have less than two-thirds ensemble agreement on whether the RR is greater or less than one. Oceans, glaciers, and bare land are masked in white.” (Lines 596-604)

Reviewer #2 (Remarks to the Author):

This is an excellent and novel study that utilizes climate model large ensembles, with and without a variety of anthropogenic forcings, to attribute projected changes in wildfire risk. The conclusions are interesting and useful, highlighting in particular the role aerosols have played in a number of regions in suppressing wildfire risk, and demonstrating impact of the future decline in that suppression effect. This is an important study, and I recommend acceptance subject to the authors addressing a few minor comments, primarily providing a few more details for the broad audience this paper will be of interest to.

We thank the reviewer for the positive comments on our manuscript.

Specific comments:

Line 93: The paper talks about "extreme" levels of wildfire risk, and chooses the 90th percentile of Canadian FWI as a criteria to define "extreme" fire behaviour. However, at a 90th percentile, on average that mean 36 days per year, one in ten days, would exceed this threshold. A 90th percentile threshold certainly describes weather suitable for fire spread, but events described as "extreme" might be expected to occur on fewer than 1 in 10 days a year. Other studies have utilized more stringent fire weather thresholds of 95 or 99% eg. see:

de Jong, M. C., Wooster, M. J., & McCall, F. F. (2016). Calibration and evaluation of the Canadian Forest Fire Weather Index (FWI) System for improved wildland fire danger rating in the United Kingdom. *Natural Hazards and Earth System Sciences*, 16(5), 1217.

Dowdy, A. J., Mills, G. A., Finkele, K., & de Groot, W. (2009). Australian fire weather as represented by the McArthur forest fire danger index and the Canadian forest fire weather index. *Centre for Australian Weather and Climate Research Tech. Rep*, 10, 91.

Arca, B., Pellizzaro, G., Duce, P., Salis, M., Bacciu, V., Spano, D., ... & Finney, M. (2010). Climate change impact on fire probability and severity in Mediterranean areas. In *In: Viegas, DX, ed. Proceedings of the VI International Conference on Forest Fire Research; 15-18 November 2010; Coimbra, Portugal. Coimbra, Portugal: University of Coimbra.*

Fréjaville, T., & Curt, T. (2015). Spatiotemporal patterns of changes in fire regime and climate: defining the pyroclimates of south-eastern France (Mediterranean Basin). *Climatic Change*, 129(1-2), 239-251.

I'm not suggesting the authors repeat the analysis with a stricter FWI threshold, rather that their analysis reflects more high fire danger conditions, rather than "extreme".

We agree with the reviewer that the 90th percentile may represent high or elevated fire weather conditions, but perhaps not "extreme" as in the other papers specified by the reviewer. We have updated our analysis to use the 95th percentile of the FWI, which is a stricter threshold that some of these studies have referred to as "extreme". We have updated the Methods to reflect this new threshold:

"For each grid point, we calculate the 95th percentile of the daily FWI across all 35 members of the ALL ensemble for every 30-year moving window from 1920-2080 ($p95_t$). $P95_t$ is the threshold used to define "extreme" fire weather for each window, t . We calculate the probability of FWI exceedance above this threshold for each member, i , of the ALL ensemble,

$$P_{i,t}^{ALL} = P(FWI_{i,t}^{ALL} \geq p95_t), \quad (1)$$

and each member, j , of the X ensemble,

$$P_{j,t}^X = P(FWI_{j,t}^X \geq p95_t), \quad (2)$$

for each 30-year moving window, t ." (Lines 315-321)

We have updated Figures 1, 2, 3, and 4, and Supplementary Figures 1, 2, 4, 8, 10, and 11 to reflect this new threshold. While our main conclusions remain unchanged, we have also updated the results section in the following locations to reflect the results using the new threshold:

“Extreme levels of FWI (exceeding the 95th percentile; see Methods--hereinafter extreme fire weather)...” (Lines 88-89)

“By 2080, greenhouse gases are expected to increase the risk of extreme fire weather by at least 50% in western North America, equatorial Africa, Southeast Asia and Australia (Figures 1h and 2a,e,g,h) and at least double this risk in the Mediterranean, southern Africa, eastern North America and the Amazon (Figures 1h and 2b,c,d,f).” (Lines 201-106)

“Most notably, in parts of the Amazon, projected greenhouse gas emissions increase extreme fire weather risk by >7 times in 2070-2080 (Figures 1h and 2c).” (Lines 106-107)

“We have performed “time of emergence” (TOE) calculations to determine when forced changes to extreme fire weather emerge from the background of its historic variability; this occurs before 2080 for 74% of the global land area (Figure 4a; see Methods), consistent with previous TOE studies using global climate models¹⁵.” (Lines 169-172)

Line 119-124: This paper will attract broad interest, it would be useful if, in a number of places, the authors provide more explanation or relevant citations of the mechanisms behind the various forcings. In this section, aerosols are described as increasing fire risk in some areas, but decreasing in others, without clear discussion of the different mechanisms acting in each case. Similarly in lines 168-169; aerosols are described as suppressing precipitation, assuming the reader understands the meteorological process by which that takes place - why would aerosols be expected to decrease precipitation in this case and not decrease it? Useful reference may be:

Alizadeh-Choobari, O. (2018). Impact of aerosol number concentration on precipitation under different precipitation rates. *Meteorological Applications*, 25(4), 596-605.

We thank the reviewer for this suggestion. We have added text in the Results and Discussion to explain the mechanisms that lead to changes in extreme fire weather conditions and associated variables under aerosol forcing.

“Over Southeast Asia, where aerosol emissions are projected to continue, the cooling effect of aerosols weakens the East Asian monsoon circulation, bringing drier conditions³³, and causing a net increase in extreme fire weather conditions. Interestingly, biomass burning aerosols have an opposite effect to industrial aerosols in some regions. For example, the Amazon region experiences an increase in extreme fire weather risk in the early 21st century due to biomass burning, possibly linked to inhibited cloud formation, which reduces low-intensity precipitation and increases temperatures^{34,35}.” (Lines 209-215)

Line 411: There is an error in this reference, the title is repeated twice.

Thank you for pointing out this error. We have corrected it.

- Reviewer: Dr Grant J Williamson

Reviewer Comments, second round

-

Reviewer #1 (Remarks to the Author):

I thank the authors for addressing all my comments.

Reviewer #2 (Remarks to the Author):

I believe the authors have adequately addressed the issues raised in review, and recommend this manuscript for publication.